# The Interplay between Oxidative Stress and the Nuclear Lamina Contributes to Laminopathies and Age-Related Diseases

**DOI:** 10.3390/cells12091234

**Published:** 2023-04-25

**Authors:** Lidya Kristiani, Youngjo Kim

**Affiliations:** 1Department of Biomedicine, School of Life Science, Indonesia International Institute for Life Science, Jakarta 13210, Indonesia; lidya.kristiani@i3l.ac.id; 2Department of Integrated Biomedical Science, Soonchunhyang Institute of Medi-Bioscience, Soonchunhyang University, Cheonan 31151, Republic of Korea

**Keywords:** lamins, laminopathy, HGPS, oxidative stress, DNA damage, senescence, aging

## Abstract

Oxidative stress is a physiological condition that arises when there is an imbalance between the production of reactive oxygen species (ROS) and the ability of cells to neutralize them. ROS can damage cellular macromolecules, including lipids, proteins, and DNA, leading to cellular senescence and physiological aging. The nuclear lamina (NL) is a meshwork of intermediate filaments that provides structural support to the nucleus and plays crucial roles in various nuclear functions, such as DNA replication and transcription. Emerging evidence suggests that oxidative stress disrupts the integrity and function of the NL, leading to dysregulation of gene expression, DNA damage, and cellular senescence. This review highlights the current understanding of the interplay between oxidative stress and the NL, along with its implications for human health. Specifically, elucidation of the mechanisms underlying the interplay between oxidative stress and the NL is essential for the development of effective treatments for laminopathies and age-related diseases.

## 1. Introduction

Reactive oxygen species (ROS) are highly reactive chemicals that originate from oxygen molecules; examples of ROS include hydroxyl radicals, peroxides, and superoxides. In living organisms, the mitochondria, peroxisomes, and chloroplasts, where respiration and photosynthesis occur, are the major sources of endogenous ROS [1]. In the mitochondria, ROS are continuously generated as byproducts of oxidative phosphorylation [1]. ROS are also generated from external sources, such as air pollutants, smoking, alcohol consumption, cooking, and radiation. Mammalian cells have developed several antioxidant defense mechanisms to eliminate excessive ROS and prevent ROS-mediated damage. These antioxidant defense systems include ROS-defusing enzymes, such as superoxide dismutase (SOD), glutathione peroxidase (GPX), and catalase (CAT), and non-enzymatic systems involving glutathione (GSH) and vitamins (A, C, and E) [2]. The protective response to excessive ROS also involves cell cycle arrest, which allows time for cells to repair DNA damage, regenerate oxidized proteins, and restore redox balance [3]. Cells also compensate for excessive ROS by altering gene expression [4].

Prolonged exposure to excessive ROS eventually leads to irreversible cellular damage, referred to as oxidative stress, via the oxidation of biomolecules such as DNA, proteins, and lipids [5]. In the normal state, cellular generation and removal of ROS are well balanced so that cells are not damaged by excessive ROS. Oxidation–reduction (redox) equilibrium is an essential component of homeostatic regulation in organisms that undergo oxidative phosphorylation. Increased ROS production without sufficient detoxification by the antioxidant defense system is attributed to a range of abnormal cellular processes, including lipid peroxidation [5], deformation of the nuclear envelope [2,6], interference with the DNA repair process, and subsequent genomic instability [7].

Aging is characterized by a gradual decline in cell and tissue functions and a higher susceptibility to internal and external stress. Various factors, including oxidative stress, telomere shortening, depletion of stem cells, mitochondrial abnormalities, glycation, immune system disorders, alterations in nuclear architecture, and subsequent epigenetic changes, have been implicated in the aging process. According to the oxidative stress theory of aging, the accumulation of cellular damage caused by an imbalance between the generation and removal of ROS is a leading cause of aging [8].

The nuclear lamina (NL) is a mesh-like protein network that primarily provides structural support in the nucleus. It is physically linked to chromatin and the cytoskeleton via the LINC complex, which is associated with their functions in essential nuclear and cytoplasmic processes, such as DNA replication, transcription, repair, and nuclear migration [9]. Mutations of NL components are tightly linked with numerous human diseases. Furthermore, recent studies have suggested that the interplay between oxidative stress and the NL plays a critical role in aging and age-related diseases. Hutchinson–Gilford progeria syndrome (HGPS), one of the most severe premature aging syndromes, is caused by mutations in *LMNA*, which encodes lamin A/C [10]. Physiological reduction in lamin B1 expression has also been reported to be associated with aging and aging-associated dysfunction in multiple tissues, whereas loss-of-function mutations in the genes encoding B-type lamins lead to lethality right after birth [11,12,13,14]. Lamin A/C is found in both the nucleoplasm and nuclear membrane because it is soluble in its mature form, while B-type lamins and progerin, the mutated form of lamin A, are always attached to the inner nuclear membrane due to their farnesyl group [15]. Numerous studies have attempted to elucidate the molecular mechanisms underlying age-related abnormalities in individuals with *LMNA* mutations. Cellular damage due to oxidative stress is one of the possible mechanisms by which *LMNA* mutations contribute to aging and aging-related diseases [16,17]; of course, it is unlikely that this is the only mechanism involved. However, most studies do not address whether *LMNA* specifically functions in this way, or if NL as a whole is critical to the maintenance of cellular functions under oxidative stress. In particular, the critical role of lamin B1 in oxidative stress-mediated cellular damage in the context of aging is underestimated. In this review, we discuss recent progress in understanding the link between oxidative stress and the NL and explore how it contributes to aging and aging-related diseases.

## 2. The Nuclear Lamina

In eukaryotes, the nuclear envelope (NE) provides a physical boundary between the cytoplasm and the nucleus. The interior surface of the NE is lined by the NL, a meshwork of fibrous proteins and their associated proteins, which are physically connected to nuclear pore complexes and perinuclear membrane space proteins [9]. The NL mostly comprises nuclear intermediate filaments (IFs) formed by lamins and provides structural integrity to the nucleus. Lamin monomers are assembled to form 3.5-nm-diameter filaments, which are then further packed to form a 14 ± 2-nm-thick layer [18].

Lamins are classified into A- and B-types based on their biochemical and immunological properties [19]. There are three lamin genes, *LMNA, LMNB1,* and *LMNB2*, in mammals. A-type lamins, such as lamins A and C, are encoded by *LMNA*. B-type lamins are encoded by two different genes; *LMNB1* encodes lamin B1 and *LMNB2* encodes lamin B2 and B3. B-type lamins are predominantly localized in the nuclear periphery, whereas lamins A and C are found in both the nucleoplasm and NE. While A-type lamins are robustly expressed in multiple adult somatic tissues, they are relatively scarce in embryonic stem cells and other cell types during the initial stages of development [20,21]. However, recent studies indicate that A-type lamins in embryonic stem cells have crucial functions in safeguarding cell fate choices [22,23]. B-type lamins are ubiquitously expressed in almost all cell types during their development; therefore, they play major structural roles in most cell types [9]. While individual filaments consist of a single lamin isoform, different types of lamin filaments are intricately organized to form a complex interwoven meshwork within the nucleus [18,24]. A proteomic analysis of different cell types from various tissues revealed that the lamin A:B ratio determines tissue rigidity, behavior, and stress resistance [25].

Since the discovery that HGPS is caused by mutations in *LMNA*, alterations in the NL components and structure have been proposed as causes of cellular and physiological aging [26,27]. Studies have shown that a decline in lamin B1 underlies age-associated loss of function in many tissues and organs [13]. Therefore, it is pertinent to discuss the interplay between the NL and oxidative stress in transmitting aging signals to downstream agents.

## 3. Oxidative Stress and Laminopathies Caused by *LMNA* Mutations

Mutations in the genes that encode for lamins, predominantly lamin A/C, cause a wide spectrum of human diseases, referred to as laminopathies, including muscular dystrophy, lipodystrophy, and systemic premature aging syndrome. HGPS, one of the most severe laminopathies, is a rare genetic disorder characterized by multisystem abnormalities, including premature aging. The most frequent mutation causing HGPS is c.1842C>T (p.G608G) in exon 11 of *LMNA*, resulting in cryptic splicing between an abnormal donor site in the middle of exon 11 and the usual acceptor of exon 12 [10]. This change causes a 50-amino acid deletion in the carboxyl-terminal tail of prelamin A, producing a truncated protein referred to as progerin. The 50 missing amino acids include the recognition sites for the prelamin A-cleaving enzyme ZMPSTE24. Consequently, progerin is normally farnesylated but cannot be further processed because of the lack of docking sites for ZMPSTE24. The farnesylated domain of progerin is firmly anchored to the nuclear membrane, leading to nuclear deformation and deleterious effects in HGPS cells [28]. Blocking the farnesylation of progerin with a farnesyltransferase inhibitor (FTI) successfully reduced the cytotoxic effects of progerin in vitro; however, a clinical trial of FTIs did not yield promising results [29]. A study showed that the interaction between progerin and wild-type lamin A/C was also a critical cause of nuclear deformation in HGPS and normal aging cells, providing a new therapeutic target for HGPS [30].

Progerin expression in various cell types induces excessive ROS production and reduces the activities of the antioxidant system [31,32,33]. Oxidative stress is also implicated in other types of laminopathies, such as Dunnigan-type familial partial lipodystrophy (FPLD), amyotrophic quadricipital syndrome with cardiac involvement, autosomal dominant Emery–Dreifuss muscular dystrophy (AD-EDMD), and restrictive dermopathy (RD) [32,33,34,35,36].

### 3.1. Effects of Excessive ROS

The downstream effects of oxidative stress, particularly those induced by progerin or other types of mutant lamin A, are characterized by persistent DNA damage and telomere shortening [37,38,39]. Fibroblasts from HGPS patients have shown higher ROS levels than those from age-matched controls [33]. Many studies have shown that fibroblasts with HGPS exhibit persistent H2AX foci [40], increased phospho-H2AX foci [41], altered H3 methylation, and HP1 downregulation, which are indicative of the accumulation of unrepaired DNA damage [4,27,42,43]. *Zmpste24^−/−^* mouse embryonic fibroblasts (MEFs) accumulate farnesylated prelamin A, exhibit increased DNA damage, and are more sensitive to DNA-damaging agents [40]. Bone marrow cells from *Zmpste24*-deficient mice show increased aneuploidy. *Zmpste24^−/−^* MEFs and HGPS fibroblasts display impaired DNA damage responses characterized by delayed recruitment of DNA damage proteins Rad50 and 53BP1 (p53-binding protein 1) to the damage sites. The presence of progerin also triggers the loss of components of the NuRD complex, which contributes to persistent DNA damage and genomic instability [41]. Interestingly, the basal levels of double-strand breaks (DSBs) in HGPS fibroblasts were reduced upon treatment with ROS scavengers, such as N-acetyl cysteine (NAC) [4] and sulforaphane [44]. Additionally, treatment with antioxidants, such as all-trans retinoic acid (ATRA), rescued cellular dynamics and proliferation in HGPS fibroblasts via recovery of the DNA damage response factor PARP1 [45]. Together, these results indicate that the interplay between mutant lamin A and oxidative stress leads to persistent cellular DNA damage.

In most somatic cells, telomere shortening is caused by decreased telomerase activity. It has been proposed that oxidative stress causes telomere shortening. In a study of fibroblasts isolated from humans and sheep, ROS accumulation and the telomere shortening rate showed a continuous, exponential correlation, regardless of the species [39]. It has also been shown that telomerases are exported from the nucleus to the cytosol upon excessive ROS accumulation [46]. Telomere shortening causes ROS accumulation, and it is well known that telomere shortening activates the p53 tumor suppressor to progressively induce cell cycle arrest, senescence, and apoptosis in response to genotoxic stimuli. Activation of p53 represses PGC-1α and PGC-1β genes on their promoters [47]. Since PGC-1α and PGC-1β are master regulators of mitochondrial physiology and metabolism, telomere shortening causes defects in mitochondrial biogenesis and function and the consequent accumulation of ROS. Furthermore, the telomere shortening rate correlates with the accumulation of DNA damage [48]. Overexpression of mutant *LMNA* variants in human fibroblasts led to accelerated telomere shortening and cellular senescence [49]. In the study, the overexpression of wild-type *LMNA* also resulted in increased telomere shortening. Oxidative stress also affects the lamin structure. Conserved cysteine residues in mammalian lamins A/C are required for cellular defense against ROS [32]. ROS can oxidize these residues and perturb their function, rendering them vulnerable to ROS-mediated damage. Therefore, it is plausible that changes in the nuclear architecture, due to the expression of either mutant or wild-type *LMNA*, initiate a reinforcing loop consisting of telomere shortening, ROS accumulation, and DNA damage.

### 3.2. Generation of Excessive ROS by Alterations in Lamin A/C

Mitochondrial dysfunction is a major source of ROS in laminopathies. Sustained accumulation of farnesylated prelamin A or depletion of normal lamin A/C results in mitochondrial dysfunction and oxidative stress in human fibroblasts [50]. The expression of cytochrome c oxidase subunit II (COX 2) is drastically decreased in human fibroblasts and adipocytes bearing lamin A variants, which lowers mitochondrial membrane potential and generates excessive ROS [34]. *Lmna*-null MEFs and human HGPS fibroblasts exhibit reduced NAD^+^ levels, unstable mitochondrial DNA, and weakened bioenergetics [51]. This dysfunction is linked to reduced PGC-1α levels, and diminished expression of NAD^+^ biosynthesis enzyme NAMPT and NAD^+^-dependent deacetylase SIRT1. This study also showed that lamin A/C aberrations lead to high levels of PARylation, which further lowers the NAD^+^ pool and contributes to impaired DNA base excision repair under oxidative stress. Impaired nuclear transport is another source of excessive ROS production in laminopathies. A study showed that the altered sensitivity of Ran to large cargo transport across the nuclear membrane contributed to disease-associated phenotypes in HGPS fibroblasts [52]. Another study found that in *Drosophila*, certain transcripts, such as those encoding the mitochondrial assembly regulatory factor (Marf) and a mitochondrial fusion factor (mitofusin), exit the nucleus through NE budding [53]. However, abnormal lamina organization caused by lamin C mutation inhibits the release of these RNAs via NE budding, leading to impaired RNA export and progressive deterioration of mitochondrial integrity, ultimately resulting in premature aging.

### 3.3. Defects in the Antioxidant System in Laminopathies

The expression levels of primary antioxidant enzymes are altered in laminopathies. Catalase (CAT) and glutathione peroxidase (GPX) activities in HGPS fibroblasts were only 50% and 30%, respectively, compared to those in normal fibroblasts [54]. However, other studies have reported elevated levels of superoxide dismutase (SOD), CAT, and glutathione (GSH)-S-transferase (GST) in fibroblasts isolated from patients with *Lmna^Y259X/Y259X^* [32,55].

Peroxisomes are subcellular organelles surrounded by membranes in which many catalytic enzymes play critical roles in scavenging ROS. Proteins targeting peroxisomes were found to be defective in fibroblasts isolated from patients with HGPS [56]. In the study, aberrant peroxisomes referred to as “peroxisome ghosts”, which lacked some of the matrix proteins, were abundant in HGPS fibroblasts. In addition, CAT activity was reduced, and consequently, high levels of ROS were observed in fibroblasts expressing progerin. Therefore, the presence of progerin sequentially leads to impaired peroxisome protein targeting, defective peroxisome assembly, reduced activity of ROS-defusing enzymes, particularly CAT, and high levels of ROS, which contribute to premature senescence phenotypes. Mature erythrocytes are constantly exposed to oxidative stress more frequently than other cell types and therefore have a highly efficient antioxidant system, particularly related to GSH. In erythrocytes, reduced GSH expression plays a prominent role in mitigating the consequent damage [57]. In a rat model of HGPS, erythrocytes showed impaired membrane integrity and oxidation of transport proteins in the presence of progerin, which decreased L-cysteine influx [58]. The decreased L-cysteine influx was attributed to the decreased erythrocyte GSH content observed in HGPS rats.

### 3.4. The Emerging Role of the Nrf2 Pathway in HGPS and Normal Aging

Nuclear factor erythroid 2-related factor 2 (Nrf2) is the master transcription factor for cytoprotective genes against oxidative stress [59]. Under normal conditions, Nrf2 is recognized by Keap1, the substrate recognition subunit (SRS) of Cul3^Keap1^ E3 ubiquitin ligase, is ubiquitinated, and is finally degraded by the 26S proteasome. However, when cells are exposed to oxidative stress, Keap1 is inactivated and Nrf2 is simultaneously released from the Cul3^Keap1^ E3 ubiquitin ligase (Figure 1). Free Nrf2 translocates to the nucleus, where it binds to the antioxidant response elements (AREs) of genes involved in antioxidant defense and detoxification pathways and activates their expression. This activation process helps alleviate cellular damage caused by oxidative stress and sustains cellular health. The downstream targets of Nrf2 include genes involved in GSH production and regulation, ROS and xenobiotic detoxification, thioredoxin-based antioxidant systems, NADPH regeneration, and heme and ion metabolism.

High-throughput screening identified CAND1, a negative regulator of Nrf2, as an effector of aging defects in HGPS [60]. The study showed that Nrf2 transcriptional activity was impaired in HGPS, and reactivation of Nrf2 reversed these defects in HGPS fibroblasts. In contrast, Nrf2 knockdown induced oxidative stress and HGPS-like defects in wild-type fibroblasts. Therefore, the downregulation of the Nrf2 pathway and consequent elevation of ROS levels may be the key mechanisms that induce various defects in HGPS. An investigation into the molecular mechanism showed that progerin has a 2–3-fold higher affinity for Nrf2 compared to wild-type lamin A in both in vivo and in vitro assays. Microscopic observation revealed that Nrf2 colocalized with progerin but not with wild-type lamin A at the nuclear periphery, suggesting its sequestration by progerin. Therefore, it has been postulated that progerin-sequestered Nrf2, particularly at the nuclear periphery, cannot bind to ARS motifs in its target genes, which might impair the cellular defense system against oxidative stress in HGPS and during normal aging.

In addition, wild-type lamin A interacts with sirtuin 6 (SIRT6), which is implicated in multiple molecular pathways, including DNA repair, telomere maintenance, and anti-inflammatory pathways [61]. Wild-type lamin A, but not progerin, enhances the deacetylase activity and efficient chromatin localization of SIRT6 to promote its functions, including DNA repair. Consistent with this finding, SIRT6 activity is defective in HGPS fibroblasts. Therefore, in the presence of progerin, wild-type lamin A may not efficiently activate or localize SIRT6 to DNA lesions in the chromatin. Moreover, several studies have shown that SIRT6 promotes the Nrf2 target gene expression [62,63]. A recent study found a strong association between progerin and ER stress in the etiology of HGPS [64]. It has been also shown that alteration in the NL causes NF-κB activation and induces proinflammatory cytokines [65]. Nrf2 plays a crucial role in ER stress and NF-κB pathways [16]. Progerin and Nrf2 are interconnected and impair the formation of functional complexes that promote various cellular defense systems against oxidative stress.

## 4. Lamin B1 Is Implicated in Cellular Senescence and Aging

In addition to lamin A, growing evidence has revealed that the function of lamin B1 is strongly associated with cellular senescence and physiological aging. Studies on B-type lamins showed that lamin B1 levels drastically decreased in cells undergoing replicative and oncogene-induced senescence [66,67,68]. Conversely, primary MEFs isolated from *Lmnb1*^Δ/Δ^ mice reached the proliferative crisis earlier than controls, suggesting that loss of lamin B1 causes replicative senescence [14]. Lamin B1 knockdown in human dermal fibroblasts also slowed cell proliferation and induced premature cellular senescence [66,68]. Interestingly, both the p53 and Rb pathways were required for lamin B1 knockdown-induced senescence, but only the p53 pathway was required for slow proliferation. Octamer transcription factor 1 (Oct-1) is a negative regulator of some antioxidant genes and is sequestered by lamin B1 in the NL [6]. In *Lmnb1*^Δ/Δ^ cells, Oct-1 is released from lamin B1 and binds to the octamer sequences of genes against oxidative stress in the nucleoplasm, which results in the repression of those genes and the elevation of ROS levels. Consistently, the expression levels of antioxidant genes increased in *Oct-1^−/−^* cells. In addition to ROS, lamin B1 knockout-induced senescent cells are characterized by increased proinflammatory cytokine secretion, abnormal nuclear architecture, and chromatin reorganization, which are observed in most senescent cells. Therefore, it has been proposed that the loss of lamin B1 is a hallmark of cellular senescence [67]. Several studies have reported that lamin B1 accumulation results in cellular senescence, which contradicts the majority of other observations [66,69]. These seemingly opposing findings indicate that the nuclear response to oxidative stress is complex and not straightforward. However, the results suggest that changes in the composition of the NL can lead to abnormalities in the assembly and function of lamin filaments. Although the loss of lamin B1 predominates in most senescent cells, it is possible that changes in the stoichiometry of NL composition can also contribute to cellular senescence. Consistent with this, in vivo studies have shown that lamin B1 levels decrease with physiological aging in multiple organisms [11,13,70,71]. Loss of lamin B1 has also been shown to cause age-related dysfunction in some tissues [72]. However, the molecular mechanisms by which lamin B1 levels decrease in senescent and aging cells are not fully understood. Recent studies have suggested that oxidative stress may be a major contributor to decreased lamin B1 levels and vice versa in senescent cells and aging tissues.

## 5. Oxidative Stress Modulates the Expression and Stability of Lamins

Studies have shown that oxidative stress influences the expression and stability of lamins, leading to diverse effects on affected cells and tissues. Lamin expression is regulated by the tumor suppressors p53 and pRb as well as telomere shortening, which are all tightly associated with oxidative stress. The activation of p53 in response to oxidative stress triggers cell cycle arrest, senescence, and apoptosis through the transcriptional activation of its target genes. A proteomics study of HCT116, a colon carcinoma cell line, revealed that lamin A/C was upregulated by p53 activation in response to a genotoxic agent [73]. Upregulation of lamin A/C upon DNA damage was transcription-dependent and was not detected in HCT116 cells harboring the p53^−/−^ mutation, suggesting that lamin A/C is a target of p53-mediated transcriptional activation. In contrast, telomere shortening induces progerin expression in normally aging human fibroblasts [74]. As mentioned above, the expression of lamin B1, but not lamin B2, decreases under senescence conditions induced by DNA damage and oncogene expression. The expression level of lamin B1 is also reduced in cells isolated from patients with HGPS and atypical progeroid syndrome caused by other lamin A mutations [75,76]. The decrease in lamin B1 expression was pRb-dependent, suggesting that lamin B1 is a downstream target of the E2F transcription factor [68]. Therefore, the induction of oxidative stress results in an increase in lamin A expression and a decrease in lamin B1 expression via the p53 and pRb pathways.

In addition to transcriptional regulation, the stability of lamin proteins is controlled by specific degradation pathways that are closely linked to oxidative stress. Rapamycin activates autophagy by inactivating the mTOR pathway. Rapamycin treatment reduced progerin accumulation and alleviated pathogenic phenotypes in HGPS cells [77]. In addition to progerin, rapamycin-activated autophagy also degrades wild-type lamin A [78]. Conversely, excessive ROS blocks autophagy by activating the mTOR pathway during senescence and aging, which may stabilize lamin A isoforms [79]. In contrast, oxidative stress is responsible for the degradation of lamin B1. The exposure of rat fibroblasts and human carcinoma cell lines to excessive ROS results in proteasome-mediated lamin B1 degradation [80]. Prolonged DNA damage and cell cycle arrest lead to apoptosis. Both A- and B-type lamins, including lamins A, B1, and B2, are the initial targets of caspases during apoptosis. Many studies have shown that oxidative stress controls the expression and stability of lamins, and vice versa (Figure 2); however, this is not fully understood and is context-dependent.

Post-translational modifications (PTMs) in lamins play crucial roles in immature lamin processing, filament assembly, nuclear structure, and function. Oxidative stress has also been implicated in various PTMs involving lamin proteins. Conserved cysteine residues in the lamin A tail domain are hyperoxidized in response to oxidative stress, preventing inter- and intramolecular disulfide bonds in lamin A polypeptides, inducing nuclear deformation, and contributing to cellular senescence [32]. Isolated rat kidneys subjected to ischemia and reperfusion were used as models to investigate the cellular redox state in the etiology of renal diseases. Proteomic analysis of rat kidneys revealed that lamin A/C was highly oxidized under oxidative stress [81]. It is widely accepted that hyperphosphorylation of lamins by mitotic CDK is essential for NE breakdown at the onset of mitosis [82]. However, mouse embryonic stem cells lacking all lamins display fairly normal mitotic progression, suggesting that the phosphorylation of lamins is not required for the disassembly of the NL or NE breakdown in all cell types [12,83]. Studies have shown that multiple residues of lamin A/C are also phosphorylated during the interphase, and changes in phosphorylation are attributed to the etiology of laminopathy [17,84]. Oxidative stress is also linked to farnesylation of prelamin A. Farnesylated prelamin A was shown to accumulate in aging human vascular smooth muscle cells (VSMCs) [85]. Zmpste24 is a metalloprotease responsible for the cleavage of farnesylated prelamin A into mature non-farnesylated lamin A/C. The study showed that the expression level of Zmpste24 in VSMCs decreased in response to oxidative stress. Lamin B1 was also oxidized and degraded in response to oxidative stress in a human carcinoma cell line [80]. It is not fully understood how oxidative stress causes defects in the PTM of lamins and consequent functional defects. It is conceivable that alterations in lamin PTM may result in defective lamin filament assembly and nuclear deformation, which in turn cause loss of interactions with nuclear and cytoplasmic proteins and chromatin domains.

## 6. Potential Mechanisms

Thus far, we have shown that the alteration or reduction in the expression of lamins is tightly associated with oxidative stress and diverse cellular defects in laminopathies and age-related diseases. Although poorly understood, there are several non-exclusive hypotheses regarding the etiology of diseases caused by defective NL and oxidative stress.

Both the A- and B-type lamins are the most abundant proteins in the nucleus. It has been suggested that intact normal lamin filaments function as molecular sinks for ROS to protect the less abundant but critical nuclear proteins from transient and mild oxidative stress [32]. It has also been proposed that the nuclear shield, a multienzyme complex that protects the genome from toxic compounds, is present on the cytoplasmic surface of the nuclear membrane [86]. Based on this supposition, the microenvironment across the nuclear membrane is highly enriched with safeguarding machinery formed by cytosolic detoxifying and antioxidant enzymes. Indeed, the concentrations of CAT, GST, and peroxidase in the NE were several-fold higher than those in the cytosol. Nuclear deformation may disrupt the nuclear shield, which in turn leads to increased sensitivity to oxidative stress [16,87]. It has also been proposed that defective lamin filaments disrupt subcellular compartmentalization, which in turn causes transient intermixing of the nuclear and cytoplasmic contents [55]. Exposure of genomic DNA to cytoplasmic components, such as the mitochondria, increases the risk of oxidative stress and DNA damage.

Chromatin landscape and 3D genome organization are critical mechanisms in cellular senescence and age-associated diseases [88]. Lamins have been shown to maintain the chromatin structure. Lamin B1-depleted senescent cells displayed global changes in gene expression and chromatin state, which were similarly observed in HGPS cells [89]. Hi-C mapping revealed that disruption of the NL either by progerin expression or lamin depletion leads to global 3D genome organization changes [90,91]. Disrupted lamina–chromatin interactions can cause mislocalization of heterochromatin domains and their internal chromatin interactions, which in turn cause aberrant global gene expression profiles. While oxidative stress can disrupt the lamina, leading to aberrant lamina–chromatin interactions, lamina–chromatin interactions also regulate oxidative stress by influencing the expression of genes implicated in oxidative stress.

## 7. Conclusions

The complex and dynamic interplay between oxidative stress and NL has been the subject of extensive research in recent years. The interaction between oxidative stress and the NL is bidirectional. Oxidative stress damages the NL, leading to structural and functional changes. Defects in the NL caused by genetic mutations or aging can make it more susceptible to oxidative stress. This bidirectional relationship contributes to the development of laminopathies. Additionally, oxidative stress plays a role in age-related diseases, which are characterized by a progressive decline in cellular function. Understanding the mechanisms governing the interplay between oxidative stress and the NL is essential for developing effective treatments for laminopathies and age-related diseases. Potential therapeutic targets include reducing oxidative stress levels and protecting the NL from oxidative stress. However, to minimize the adverse effects of targeting upstream pathways, it is critical to identify the specific targets of oxidative stress and defective NL, which are responsible for laminopathies and age-related diseases. In this regard, it is attractive to explore the 3D chromatin structure of laminopathy or aging cells and to identify specific long-range chromatin interactions involved in the expression of antioxidant genes. It is also essential to clarify if specific isoforms of lamins are critical for cellular functions in different cell types and tissues, or if the integrity of the entire NL is ubiquitously required to ensure cellular health in all cell types. Studies with complete lamin null cells or animals might provide an opportunity to address the above challenging question. Overall, studying the interplay between oxidative stress and the NL is an exciting and rapidly evolving field with the potential to yield significant progress in public health.

## Figures and Tables

**Figure 1 cells-12-01234-f001:**
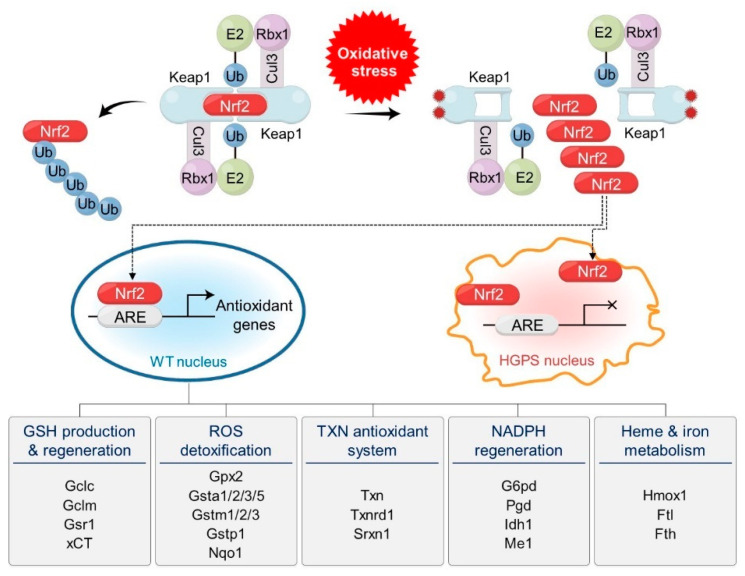
Oxidative stress and progerin are implicated in the regulation of Nrf2, the master transcription factor for cellular defense systems against oxidative stress. Dimeric Cul3^Keap1^ E3 ubiquitin ligase complexes target Nrf2 for polyubiquitination and degradation under normal conditions. Under oxidative stress, cysteine residues of Keap1 are oxidized and the structure of Nrf2-binding motifs is modified, which releases Nrf2 from Cul3 ligase complexes. Released Nrf2 translocates into the nucleus and activates antioxidant genes by binding to antioxidant response elements (AREs) in normal cells. Over 200 antioxidant genes are activated by Nrf2. They are classified by their functions and include genes involved in GSH production and regeneration, ROS detoxification, TXN antioxidant systems, NADPH regeneration, and heme and iron metabolism. Representative genes are indicated under the functions. In HGPS cells, progerin sequesters Nrf2 at the nuclear periphery, which prevents the activation of antioxidant genes.

**Figure 2 cells-12-01234-f002:**
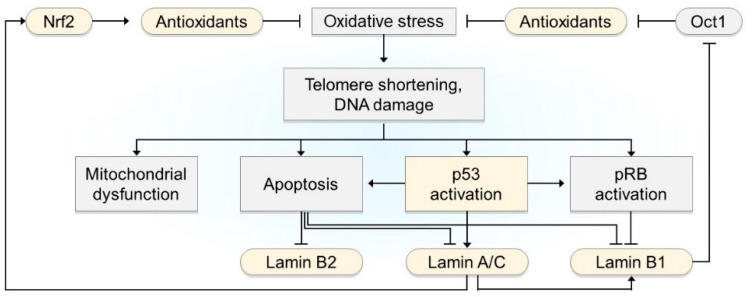
A summary of the major pathways that regulate oxidative stress and lamins. Transient oxidative stress activates p53, which induces an increase in lamin A/C and pRB expression, which causes a decrease in lamin B1 expression. Prolonged oxidative stress and DNA damage cause apoptosis, which cleaves all lamins, including lamin B2. Oxidative stress-mediated telomere shortening interferes with mitochondrial biogenesis and function, generating more ROS. Lamins also regulate oxidative stress. Wild-type lamin A/C allows Nrf2 binding to the ARS of antioxidant genes against oxidative stress, but progerin sequesters Nrf2, preventing the activation of antioxidant genes. Lamin B1 sequesters Oct1, a repressor of antioxidant genes, under normal conditions. During oxidative stress, reduced levels of lamin B1 cause the release of Oct1, leading to the suppression of antioxidant genes. Negative regulators of oxidative stress are highlighted in yellow, and positive mediators of oxidative stress are highlighted in gray. Genes are represented by rounded rectangles, while cellular processes are depicted using rectangular shapes.

## Data Availability

No new data were created or analyzed in this study. Data sharing is not applicable to this article.

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
