# Peer review of "The Interplay between Oxidative Stress and the Nuclear Lamina Contributes to Laminopathies and Age-Related Diseases"

_cells, 2023, doi:10.3390/cells12091234_

Round 1
Reviewer 1 Report
The review by Lidya Kristiani and Youngjo Kim provides a good overview of the interplay between oxidative stress and the nuclear lamina in aging. The topic is exciting and important. However, there are some minor points for improvement.
1. Line 27 ... endogenous ROS. Please add a citation
2. Line 57 ... and nuclear migration. Please add a citation
3. Line 65 … lead to embryonic lethality. Lmnb1-/- and Lmnb2-/- die after birth (Vergnes, Péterfy et al. 2004, Kim, Sharov et al. 2011). Please correct.
4. Line 91 A-type lamins are also expressed in mouse embryonic stem cells and play critical roles in safeguarding cell fate choices (Guo, Kim et al. 2014, Amendola and van Steensel 2015, Wang, Elsherbiny et al. 2022). Please correct.
5. Line 133, H2AX foci not H2AC foci
6. Please extend and discuss the relationship between mitochondrial dysfunction, oxidative stress, and LMNA mutations in section 3.2.
8. In Section 4, it is important to note that the relationship between lamin B1 and oxidative stress is complicated. On the one hand, oxidative stress induces senescence via an increase in lamin B1 expression; on the other, both overexpression and depletion of lamin B1 lead to a decrease in ROS, which might involve different pathways. Please add the following references:
Barascu A, Le Chalony C, Pennarun G, et al. Oxidative stress induces an ATM‐independent senescence pathway through p38 MAPK‐mediated lamin B1 accumulation[J]. The EMBO journal, 2012, 31(5): 1080-1094.
Freund A, Laberge R M, Demaria M, et al. Lamin B1 loss is a senescence-associated biomarker[J]. Molecular biology of the cell, 2012, 23(11): 2066-2075.
Shimi T, Butin-Israeli V, Adam S A, et al. The role of nuclear lamin B1 in cell proliferation and senescence[J]. Genes & development, 2011, 25(24): 2579-2593.
Dysregulated mitochondrial integrity associated with lamin B1 reduction is involved in cellular senescence progression in COPD pathogenesis.
Author Response
We really appreciate the reviewer’s comments. In response to the reviewer 1’s comments, we modified the main text of manuscript and added some references. The changed parts are marked by yellow highlights in the revised manuscript.
The review by Lidya Kristiani and Youngjo Kim provides a good overview of the interplay between oxidative stress and the nuclear lamina in aging. The topic is exciting and important. However, there are some minor points for improvement.
- Line 27 ... endogenous ROS. Please add a citation
We added a reference.
- Line 57 ... and nuclear migration. Please add a citation
We added a reference.
- Line 65 … lead to embryonic lethality. Lmnb1-/- and Lmnb2-/- die after birth (Vergnes, Péterfy et al. 2004, Kim, Sharov et al. 2011). Please correct.
Sorry for this mistake. We changed this part and added references.
- Line 91 A-type lamins are also expressed in mouse embryonic stem cells and play critical roles in safeguarding cell fate choices (Guo, Kim et al. 2014, Amendola and van Steensel 2015, Wang, Elsherbiny et al. 2022). Please correct.
Our apologizes for the inaccuracy in this part. We changed this part with recent updates about lamin A/C in embryonic stem cells.
- Line 133, H2AX foci not H2AC foci
We corrected this typo.
- Please extend and discuss the relationship between mitochondrial dysfunction, oxidative stress, and LMNA mutations in section 3.2.
We extended this section as the reviewer recommended.
- In Section 4, it is important to note that the relationship between lamin B1 and oxidative stress is complicated. On the one hand, oxidative stress induces senescence via an increase in lamin B1 expression; on the other, both overexpression and depletion of lamin B1 lead to a decrease in ROS, which might involve different pathways. Please add the following references:
Barascu A, Le Chalony C, Pennarun G, et al. Oxidative stress induces an ATM‐independent senescence pathway through p38 MAPK‐mediated lamin B1 accumulation[J]. The EMBO journal, 2012, 31(5): 1080-1094.
Freund A, Laberge R M, Demaria M, et al. Lamin B1 loss is a senescence-associated biomarker[J]. Molecular biology of the cell, 2012, 23(11): 2066-2075.
Shimi T, Butin-Israeli V, Adam S A, et al. The role of nuclear lamin B1 in cell proliferation and senescence[J]. Genes & development, 2011, 25(24): 2579-2593.
Nayuta Saito, Jun Araya, Saburo Ito et al. Dysregulated mitochondrial integrity associated with lamin B1 reduction is involved in cellular senescence progression in COPD pathogenesis. European Respiratory Journal 2018 52: PA2177
We agree with reviewer that the relationship between lamin B1 and oxidative stress is not straightforward. In response to the reviewer’s critics, we modified this part in the main text of the revised manuscript and added the missing reference.

Reviewer 2 Report
This very interesting manuscript deals with relationships between lamins and oxidative levels in cells. It provides a useful and comprehensive review of literature on the subject. It is clearly written, well documented and all data are referenced.
I was very interested by the topic, and I thank the authors for their work.
I only have minor concerns before publication.
- In introduction, it may be useful to precise that A-type lamins are solubles on their mature form (but not progerin, that remains farnesylated and then, anchored to membranes how this is explained after), whereas B-type lamins always wear their farnesyl (or geranyl-geranyl) anchor, thus being stuck at the inner nuclear membrane. This is of importance to understand lamin nuclear distribution. This has been well illustrated by Lévy and colleagues https://doi.org/10.1093/hmg/ddl214, a reference that should be (and others from that team) cited.
- L110. The cryptic splicing does not occur in intron 11, but within exon 11 (3' end), leading to a 150 nucleotides shorter mRNA. Please, correct.
- L227. As far as I know, progerin, regarding its farnesylation status, remains anchored to the inner nuclear membrane, thus locates only at the nuclear periphery, not the nucleoplasm. Please correct the sentence.
- L239-240. For the same reason, it is not necessary to state that progerin sequester Nrf2 at the nuclear periphery.
- L270. Figure 2. I wonder if this illustration would become more understandable if genes and mechanisms were put in differently shaped and/or colored boxes ?
- L326. Figure 2 legend. This seems paradoxal : if lamin B1 level decreases upon oxidative stress, thus it is counterproductive with sequestration of Oct1, which will be less effective ??? I'm not sure to perfectly understand this idea.
Again and apart from these minor things, I thank once again the authors for their nice work.
Author Response
We really appreciate the reviewer’s comments. In response to the reviewer 2’s comments, we modified the main text and figure legend of the manuscript. The changed parts are marked by sky blue highlights in the revised manuscript.
This very interesting manuscript deals with relationships between lamins and oxidative levels in cells. It provides a useful and comprehensive review of literature on the subject. It is clearly written, well documented and all data are referenced.
I was very interested by the topic, and I thank the authors for their work.
I only have minor concerns before publication.
- In introduction, it may be useful to precise that A-type lamins are solubles on their mature form (but not progerin, that remains farnesylated and then, anchored to membranes how this is explained after), whereas B-type lamins always wear their farnesyl (or geranyl-geranyl) anchor, thus being stuck at the inner nuclear membrane. This is of importance to understand lamin nuclear distribution. This has been well illustrated by Lévy and colleagues https://doi.org/10.1093/hmg/ddl214, a reference that should be (and others from that team) cited.
Thanks for this comment. We modified this part as reviewer recommended and added the reference.
- L110. The cryptic splicing does not occur in intron 11, but within exon 11 (3' end), leading to a 150 nucleotides shorter mRNA. Please, correct.
Sorry for inaccuracy in this part. We modified this part to ensure that splicing happens in between sites in the middle of exon 11 and the regular splicing acceptor in front of exon 12.
- L227. As far as I know, progerin, regarding its farnesylation status, remains anchored to the inner nuclear membrane, thus locates only at the nuclear periphery, not the nucleoplasm. Please correct the sentence.
We agree with reviewer’s comment. We corrected this part as reviewer recommended.
- L239-240. For the same reason, it is not necessary to state that progerin sequester Nrf2 at the nuclear periphery.
We agree with reviewer’s comment. We corrected this part as reviewer recommended.
- L270. Figure 2. I wonder if this illustration would become more understandable if genes and mechanisms were put in differently shaped and/or colored boxes ?
Thanks for this comment, which will make the figure more understandable. We modified the color and shapes of the figure 2.
- L326. Figure 2 legend. This seems paradoxal : if lamin B1 level decreases upon oxidative stress, thus it is counterproductive with sequestration of Oct1, which will be less effective ??? I'm not sure to perfectly understand this idea.
Sorry about the ambiguity of the sentences. We changed this part to make clear that Oct1 is released from lamin B1 under oxidative stress, which will repress antioxidant genes.
Again and apart from these minor things, I thank once again the authors for their nice work.
